# Analysis of Survival-Related lncRNA Landscape Identifies A Role for LINC01537 in Energy Metabolism and Lung Cancer Progression

**DOI:** 10.3390/ijms20153713

**Published:** 2019-08-01

**Authors:** Wei Gong, Lei Yang, Yuanyuan Wang, Jianfeng Xian, Fuman Qiu, Li Liu, Mingzhu Lin, Yingyi Feng, Yifeng Zhou, Jiachun Lu

**Affiliations:** 1The State Key Lab of Respiratory Disease, The institute for Chemical Carcinogenesis, Collaborative Innovation Center for Environmental Toxicity, Guangzhou Medical University, Xinzao, Panyu District, Guangzhou 511436, China; 2Department of Genetics, Medical College of Soochow University, 1 Shizi Road, Suzhou 215123, China

**Keywords:** LINC01537, PDE2A, survival, energy metabolism, lung cancer

## Abstract

Many long non-coding RNAs (lncRNAs) have emerged as good biomarkers and potential therapeutic targets for various cancers. We aimed to get a detailed understanding of the lncRNA landscape that is associated with lung cancer survival. A comparative analysis between our RNA sequencing (RNA-seq) data and TCGA datasets was conducted to reveal lncRNAs with significant correlations with lung cancer survival and then the association of the most promising lncRNA was validated in a cohort of 243 lung cancer patients. Comparing RNA-seq data with TCGA ones, 84 dysregulated lncRNAs were identified in lung cancer tissues, among which 10 lncRNAs were significantly associated with lung cancer survival. LINC01537 was the most significant one (*p* = 2.95 × 10^−6^). Validation analysis confirmed the downregulation of LINC01537 in lung cancer. LINC01537 was observed to inhibit tumor growth and metastasis. It also increased cellular sensitivity to nilotinib. PDE2A (phosphodiesterase 2A) was further identified to be a target of LINC01537 and it was seen that LINC01537 promoted PDE2A expression via RNA–RNA interaction to stabilize PDE2A mRNA and thus echoed effects of PDE2A on energy metabolism including both Warburg effect and mitochondrial respiration. Other regulators of tumor energy metabolism were also affected by LINC01537. These results elucidate a suppressed role of LINC01537 in lung cancer development involving tumor metabolic reprogramming, and we believe that it might be a biomarker for cancer survival prediction and therapy.

## 1. Introduction

Molecular targeted therapy (MTT) has been one of the most popular modes of cancer treatment for several years, which works by blocking the growth of cancer cells via interfering with specific molecules that play essential roles in cancer development. With the expectation of it being more effective and causing less harmful side effects in comparison with traditional hormonal therapy and cytotoxic chemotherapy, MTT has had such a quick development that a series of drugs such as Gefitinib, Erlotinib were developed, accompanied with a better understanding of molecular pathogenesis in human cancer [1]. However, with the development of targeted therapy, there is increasing evidence that tumor cells develop drug resistance and ultimately leading to induced treatment failure. Meanwhile, only 10–20% of cancer patients respond well to these existing targeted drugs. Another problem that is often experienced is that once the primary tumorous clones are treated by targeted drugs, the secondary clones that are not sensitive to the drugs gain a growth advantage and cause recurrence of cancer and metastasis [2]. These challenges encouraged us to explore the molecular landscape for cancer and exploit comprehensive biomarkers that can aid in development of new MTT drugs.

Recent large-scale RNA sequencing (RNA-seq) analyses have shown that more than two-thirds of the human genome are transcribed but less than 2% of them are translated into protein, which have been widely studied in association with various human diseases and some of them have become therapeutic targets [3,4]. However, the transcribed non-coding RNA (ncRNA) lack detailed researches, especially the long ncRNA (lncRNA) that represent a class of transcripts longer than 200 nucleotides with complex, diverse, and largely uncharacterized biological functions [5]. Through RNA–DNA, RNA–RNA, and RNA–protein interactions, several lncRNAs have been identified to act as mediators on regulating target coding-gene expressions on transcriptional or post-transcriptional level involving epigenetic regulation, gene imprinting, mRNA splicing, distal enhancer, and so on [6,7,8,9,10]. Abnormal expression profiles of lncRNAs revealed by an abundance of RNA-seq analyses have initially foreboded certain lncRNAs as oncogenic or tumor-suppressive roles in pathogenesis of various cancers, but there is a dearth of comprehensive studies to characterize effects of these abnormally expressed lncRNAs.

Lung cancer has been the leading cause of cancer-related death for several years, with a five-year survival rate below 20% [11,12]. Although a lot of studies have revealed several lncRNAs associated with lung cancer risk and prognosis, effects of a large proportion of lncRNAs on lung cancer development are still unexplored. Therefore, we conducted a comparative analysis of our RNA-seq data and TCGA data and discovered the survival-related lncRNA landscape for lung cancer. Among these lncRNAs, we characterized the functional role of LINC01537 in energy metabolism and lung cancer progression.

## 2. Results

### 2.1. Identification of lncRNAs Landscape for Lung Cancer Survival

RNA-seq identified 230 lncRNAs (110 upregulated and 120 downregulated) with significantly different expressions between lung cancer tissues and corresponding noncancer lung tissues (Figure 1a). Some of these lncRNAs like FENDRR (FOXF1 adjacent non-coding developmental regulatory RNA), UCA1(urothelial cancer associated 1) have been reported to play roles in lung cancer development [13,14]. There were also hundreds of differentially expressed lncRNAs in LUAD (lung adenocarcinoma) and squamous cell carcinoma (LSCC) as revealed by TCGA data analysis (Appendix A). BLAST analysis showed that 84 lncRNAs were overlapping in three differential lncRNAs expression profiles (Figure 1b, Appendix A). Further univariate cox model revealed 10 lncRNAs to be significantly associated with lung cancer survival (Figure 1c), among which LINC01537 was the most significant one (*p* = 2.95 × 10^−6^). The distribution of patient risk scores and the 10 lncRNAs expression values are shown in Figure 1d. The high-risk group indicated a significantly better survival than the low-risk group (*p* < 0.001; Figure 1e) and showed significant predictive accuracy on lung cancer survival (Figure 1f).

### 2.2. LINC01537 Is Downregulated in Lung Cancer

Public ribosome profiling revealed LINC01537 was not bound by the ribosome [15], suggesting that it could not code protein. CPATool also indicated the lncRNA having no coding probability (http://lilab.research.bcm.edu/cpat/index.php; Appendix A). As shown in Figure 1g, 72.0% (116/161) of lung cancer tissues showed a significantly decreased expression of LINC01537 when compared to paired noncancerous lung tissues in the southern samples (*p* = 0.027), which is consistent with TCGA that shows a low expression of the lncRNA in lung cancer (Appendix A) and in almost all cancer types (Appendix A). Furthermore, downregulated LINC01537 was observed in lung cancer in the eastern samples with a margin at the edge of significance (*p* = 0.063). In addition, cancer cells generally expressed lower LINC01537 than noncancer cells (Figure 1h).

### 2.3. LINC01537 Inhibits Lung Cancer Development

The qPCR analysis confirmed overexpression of LINC01537 in both A549 cells and PC9 cells in response to lentivirus infection (Appendix A). LINC01537 overexpression led to step-down cell proliferation rate (Figure 2a), low tumor formation rate (Figure 2b) in vitro, and little tumor growth in vivo (Figure 2c–e). Furthermore, LINC01537 overexpression induced lung cancer cells stop in the period of G0/G1 (Figure 2f) and accelerate cell apoptosis (Figure 2g). Moreover, overexpression of LINC01537 attenuated cell migration and invasion in vitro and in vivo (Figure 3a–e). IHC (Immunohistochemistry) confirmed that the nodules harbored prominent features of LUAD (Figure 3f).

### 2.4. Overexpressed LINC01537 Benefits Anticancer Medicament

We aimed to discover whether LINC01537 affects anticancer therapy of lung cancer. Cell growth repression mediated by nilotinib was remarkably augmented in presence of LINC01537 overexpression (Figure 4a). However, the effect was not conspicuous for cisplatin (Figure 4b).

### 2.5. LINC01537 Regulates PDE2A Expression by Stabling PDE2A mRNA

To explore functional mechanism, the subcellular localization of LINC01537 was examined, leading to the finding that LINC01537 predominately resided in the cytoplasm (Figure 5a). It was thus hypothesized that LINC01537 may target mRNA via RNA–RNA interaction as other studies have shown [16]. To identify target mRNAs that could interact with LINC01537, we next used IntraRNA to predict interactions between the lncRNA and possible target mRNAs, whose expressions were significantly correlated with LINC01537 as gene coexpression network analysis of TCGA LUAD as well LSCC data shown (Table 1). Remarkably, PDE2A (phosphodiesterase 2A) whose expression harboring the highest correlation coefficient with LINC01537 was identified to be a target (Figure 5b). PDE2A is located in ~3kb downstream of LINC01537, but they do not share promoters, giving they are transcribed in the opposite direction. We confirmed the significant correlation between expressions of PDE2A and LINC01537 in our clinical samples (*r* = 0.630, *p* < 0.001; Figure 5c) and cells (Figure 5d). It is of substantial interest that the specificity of PDE2A and LINC01537 expressions are highly consistent across different types of tissue (Figure 5e). Further ribonuclease protection assay (RPA) revealed considerably higher expression of PDE2A in LINC01537 overexpressed cells than in the control cells (Figure 5f), and mRNA stability test showed lower decreased rates of PDE2A mRNA in the former cells than the latter ones (Figure 5g). Thus, LINC01537 stabilizes PDE2A mRNA in vivo.

We wanted to understand if PDE2A was a major effector of LINC01537-mediated tumor suppression. By blocking PDE2A with dipyridamole and silencing it with small interfering RNA (siRNA) (Appendix A), cell proliferation was significantly increased in presence of dipyridamole (Figure 5h) as well as PDE2A siRNA (Appendix A), implying that PDE2A serves as an effector of LINC01537.

### 2.6. LINC01537 Affects Both the Mitochondrial Respiration and Warburg Effect

Given that PDE2A is a key regulator of mitochondrial respiration, it was inquired whether LINC01537 has a similar effect [17]. Seahorse analysis revealed decreased oxygen consumption rate (OCR) in LINC01537 overexpressed cells. The lncRNA suppressed both base respiration and ATP turnover (Figure 6a). Interestingly, the analysis also showed declined extracellular acidification rate (ECAR) including cell glycolysis, decreased glycolytic capacity, and glycolytic reverse in cells overexpressing LINC01537 (Figure 6b). To further study the potential function of LINC01537 in the regulation of energy metabolism, a panel of genes involved in it were tested. PGK1, DPYSL4, L32, GLS2GLUT1, PKM2, MCP1, and IFNg were consistently upregulated, while GLUT1 and PKM2 were downregulated in both A549 and PC9 cells overexpressing LINC01537 when compared to the control cells (Figure 6c).

## 3. Discussion

Since prodigious lncRNAs have been discovered and some of them play multitudinous and pivotal roles in lung cancer development, it is imperative to ascertain which ones are related to cancer survival, which have potential to be not only prognostic indicators but also therapeutic targets for lung cancer. Even though previous studies have revealed many lncRNAs with significant prognostic value for lung cancer [18], we are only observing the tip of the iceberg. Our comprehensive analysis of differential expression profiles of lncRNAs and followed bioinformatics analysis with TCGA data delineated a ten lncRNAs landscape for lung cancer survival, and further identified the most significant candidate LINC01537 that we studied the characteristics with the help of functional assay. We found that LINC01537 functions as a regulator of energy metabolism to suppress lung cancer development via PDE2A, underlying a mechanism that LINC01537 stabilizes PDE2A mRNA via RNA–RNA interaction (Figure 6d).

Some of the ten lung cancer survival-related lncRNAs echoes previous studies on cancer. A recent investigation reported that LINC00973 was involved in chemotherapy resistance of colon cancer [19]. CTD-2357A8.3 was identified as a biomarker for predicting survival in esophageal cancer [20]. FENDRR (FOXF1 adjacent non-coding developmental regulatory RNA) is also one of the most well-known lncRNAs that functions to inhibit tumor including lung cancer [14]. Remarkably, a lncRNA-miRNA-mRNA triple network analysis showed that RP11-613D13.8 can sponge miR-191-5p, which were confirmed to be tumor suppressive and metabolism-related in several tumors [21]. Moreover, LINC00092 can drive glycolysis and promotes ovarian cancer progression via cancer-associated fibroblast [22], further suggesting that lncRNAs involving energy metabolism play a big part in cancer survival. In addition, the TCGA data suggested a significant association between expressions of RP11-221N13.4 and LINC01537 [23]. All these factors highlight the importance of lncRNAs in cancer metabolism and further studies are warranted to expand on these findings, especially their mechanism of action.

LINC01537 is located on human chromosome 11q13.4, a locus with high frequency of copy number change in LUAD [24], which may be a reason for LINC01537 abnormal expression in lung cancer. LINC01537 acts as a tumor suppressor to prohibit tumor cell growth, migration, and invasion, by promoting apoptosis and to regulating cell cycle, further supporting the hypothesis that it could be a potential therapeutic target. To further reveal the molecular mechanism of how LINC01537 regulates cancer development, we tentatively verified that the lncRNA can increase PDE2A expression by RNA–RNA interaction. This mechanism for lncRNA has been confirmed repeatedly in other studies [25,26,27,28].

As an important member of phosphodiesterases (PDEs) family, PDE2A can regulate mitochondrial cAMP (Cyclic Adenosine monophosphate) levels and respiration, participating in various physiological activities such as energy metabolism [17]. The location of PDE2A gene is next to LINC01537, thus presenting a high probability of physical interaction between the PDE2A mRNA and the lncRNA. Interestingly, one clinical trial showed that targets on PDE2A have shown clinical benefits in 23% of patients with metastatic breast cancer [29]. In this study, it was illustrated that LINC01537 has the ability to influence a molecularly targeted agent in treating lung cancer cells. Thus, the fact that LINC01537 acts upstream of PDE2A suggests the possibility that targeting LINC01537 may provide an additional option to patients that are insensitive to PDE2A targeted drugs.

Since PDE2A plays a pivotal role in energy metabolism, there can be no doubt that LINC01537 exerted an effect on tumor energy metabolism. It can depress both the Warburg effect and mitochondrial respiration. Deregulating cellular energetics is one of the ten characteristics of cancer [30]. Despite Warburg effect being well-known for the fact that tumor cells tend to rewire their metabolism with increased glycolytic rate and fermentation of glucose to lactate, which helps the cells to grow, survive, and metastasize [31], mitochondrial respiration is still an important mediator of tumorigenesis and plays an important role in supporting tumor cell metabolism [32,33]. Thus, not only Warburg effect but also mitochondrial respiration could be enhanced due to LINC01537 deficiency and lead to a poor prognosis of lung cancer. Our data depicts a novel lncRNA involved in mitochondrial activity and metabolic reprogramming in a similar way as previously reported lncRNAs [34].

## 4. Methods

### 4.1. Patient Sample Collection

Two hundred forty-three paired cancerous and corresponding noncancer lung tissues were obtained from patients suffering from histopathologically confirmed lung cancer during the surgical excision in this study. The subjects have been described in previously published studies [30,35]. In total, 161 samples from southern China were collected from 3 hospitals in Guangzhou and Kunming city and 82 samples from eastern China were gathered from 1 hospital in Suzhou city. The demographics and clinical features of studied subjects are listed in Table 2. The study was approved by the Ethics Committee of Guangzhou Medical University and Soochow University (Permit NO. 20160315. 15 March 2016).

### 4.2. RNA-seq

8 pairs of cancerous and corresponding noncancer lung tissues were randomly selected for RNA-seq analysis on Illumina Hiseq 2500 by a commercial biotechnology company (BIOMIAO biological, Beijing, China). Read mapping was conducted with TopHa2. LncRNAs with false discovery rate (FDR) < 0.01 and |log_2_Fold change (logFC)| > 1.5 were considered to be significant.

### 4.3. TCGA Data Analysis

The RNA-seq BAM files of lung cancer were downloaded from TCGA database (https://cancergenome.nih.gov/) including 997 lung cancer samples (i.e., 506 adenocarcinomas and 491 squamous-cell carcinomas) and 107 noncancer lung samples. Differential expression profiling was performed with DESeq2 package using R programming language. The criterion of significance for lncRNAs was the same as above. Overlapped significant lncRNAs between RNA-seq result and TCGA data were further fitted in a univariate Cox regression model. A risk score was calculated based on significant lncRNAs weighted by their respective Cox regression coefficients [36]. Then TCGA patients were divided into low-risk and high-risk groups using the median risk score as the cut-off.

### 4.4. Cell Culture

A549 and PC-9 human lung cancer cell lines used in this study were purchased from Cell Bank of Type Culture Collection of the Chinese Academy of Science (Shanghai Institute of Cell Biology, Shanghai, China) on July 25, 2017. Both cell lines were tested and authenticated using a panel of genetic and epigenetic markers before being used in this study on July 31, 2017. All cells except 293T were cultured in RPMI-1640 (Gibco, CA, USA) supplemented with 10% fetal bovine serum (FBS, Gibco). The 293T cells were cultured in DMEM (Gibco) with 10% FBS. Cells were placed in a CO_2_ incubator (SANYO ElectricCo., Ltd., Osaka, Japan) with constant 90% humidity and 5% CO_2_.

### 4.5. RNA Isolation and Quantitative Real-Time RT-PCR (qPCR)

Total RNA was extracted using TRIzol reagent (Invitrogen, carlsbad, CA, USA). Expressions of LINC01537, PDE2A, a panel of energy metabolism-related genes, and GAPDH (i.e., an internal control) were determined using SYBR-green qPCR (TAKARA, Osaka, Japan). The 2^−ΔCT^ or 2^−ΔΔCT^ method was used to calculate expression level. The primes for these genes are shown in Appendix A.

### 4.6. Construction of LINC01537 Stable Overexpression Cells

Full-length cDNA of human LINC01537 was synthesized and cloned into lentiviral expression vector pEZ-Lv201 by Genecopoeia Biotech Co. Ltd. (Guangzhou, China). The pEZ-Lv201-LINC01537 was then used to produce virus particles and transfected into A549 and PC9 cell lines. Empty pEZ-Lv201.1 was used as a control. The detailed procedure for construction is presented in Appendix A.

### 4.7. Cell Phenotypic Experiments

Cell proliferation and migration measured by wound healing assay were directly determined and imaged using IncuCyte^®^ Live Cell Analysis Imaging System (Essen BioScience Co., Ltd., Ann Arbor, Michigan, USA) [37,38]. Cell cycle and apoptosis were measured by flow cytometry (FCM). Cell migration and invasion were tested by Transwell assay with uncoated or Matrigel-coated (BD Biosciences, San Jose, CA, USA) Boyden chambers and scratch test. Clonogenic ability was examined by tablet cloning assay. The detailed protocols for these experiments are presented in Appendix A.

### 4.8. Tumor Xenografts and Neoplasms Metastatic Model in Mice

Four to six-week-old female BALB/C-nu mice purchased from Huarongkang Biotechnology Co. Ltd. (Beijing, China) were raised at the Animal Experimental Center of Guangzhou Medical University. The mice were given a suspension of 2 × 10^6^ cells subcutaneously. The mice were examined 3 times a week for 3 weeks and tumor growth was assessed by measuring length and width of the tumor mass. The volume of the tumor was calculated using the following formula: volume = length × width^2^ × 0.5. Each group included four mice. In other side, mice were given a tail vein injection of 2 × 10^6^ cells and killed in six weeks after inoculation. The lungs were removed and fixed in 4% paraformaldehyde. The lung sections were stained with hematoxylin-eosin (HE) for histological examination. HE staining was performed following the routine protocol. Each group included four mice. All experiments and procedures involving animals were conducted in accordance with guidelines approved by the Laboratory Animal Center of Guangzhou Medical University (Permit NO. 20180320. 20 March 2018).

### 4.9. Drug Sensitivity Assay

Cells were treated with cisplatin in half of the maximal inhibitory concentration and tyrosine kinase inhibitor (TKI) nilotinib in mean plasma drug concentration (i.e., 3.6 μmol/L) [39]. The cell proliferation was determined as described above. The detailed protocol for IC50 (Half maximal inhibitory concentration) identification is presented in Appendix A.

### 4.10. Subcellular Localization

Subcellular localization of LINC01537 was determined by fluorescence in situ hybridization (FISH) and cytosolic/nuclear fractions test. The detailed protocols for them are presented in Appendix A.

### 4.11. Ribonuclease Protection Assay (RPA) and mRNA Stability Test

Protective effect of LINC01537 on PDE2A mRNA was determined by RPA with ribonuclease A + T and stability test with actinomycin D according to routine protocols [40]. The detailed protocols for the tests are presented in Appendix A.

### 4.12. PDE2A Interference

Four small interfering RNA (siRNA) candidates were designed to target PDE2A coding sequences by siDirect version 2.0 (http://sidirect2.rnai.jp/). The siRNA sequences are shown in Appendix A. The siRNA with highest silencing efficiency was selected for further transient transfection to knock down PDE2A expression using Lipofectamine^®^ 3000 (Invitrogen, carlsbad, CA, USA). Meanwhile, 50 μg/mL dipyridamole, a phosphodiesterase (PDE) inhibitor, was also used to block PDE2A [41].

### 4.13. Glycolysis Stress Test and Cell Mitochondria Stress Test

Extracellular acidification rate (ECAR) for assessing cell glycolysis, glycolytic capacity, and glycolytic reverse was determined using Seahorse XF Glycolysis Stress Test Kit (Agilent, Santa Clara, CA, USA) according to the manufacturer’s instructions. Oxygen consumption rate (OCR) for evaluating mitochondrial basal respiration and ATP turnover was measured with Seahorse XF Cell Mito Stress Test Kit (Agilent, Santa Clara, CA, USA) as per the manufacturer’s instructions. Cells were seeded into Seahorse XF cell culture plates and ECAR and OCR were detected in XF96 Analyzer (Agilent, Santa Clara, CA, USA).

### 4.14. Statistical Analysis

Differences of qualitative and quantitative data between two groups were evaluated by χ2 test, Student’s *t* test, and two-way ANOVA, respectively. Differences of gene expression between paired cancerous and noncancerous tissues were assessed using paired *t* test. Event-time distribution was estimated based on Kaplan–Meier method. Survival analysis was obtained by using log-rank test and Cox regression model. Predictive value was determined with time-dependent receiver operating characteristic (ROC) curves using timeROC package in R programming [42]. All tests were two-sided using Stata software (version 12.0). *p* < 0.05 was considered to be statistically significant.

## 5. Conclusions

Tens of thousands of lncRNAs have enriched the human transcriptome diversity and provided novel and meritorious targets for both prediction of cancer prognosis and molecular targeted therapy. Since LINC01537 acts as a tumor suppressor gene via interaction with PDE2A to depress tumor energy metabolism, it will be of substantial interest to use this lncRNA accompanied with other biomarkers as a tool for assessing lung cancer survival and devise drugs to target it.

## Figures and Tables

**Figure 1 ijms-20-03713-f001:**
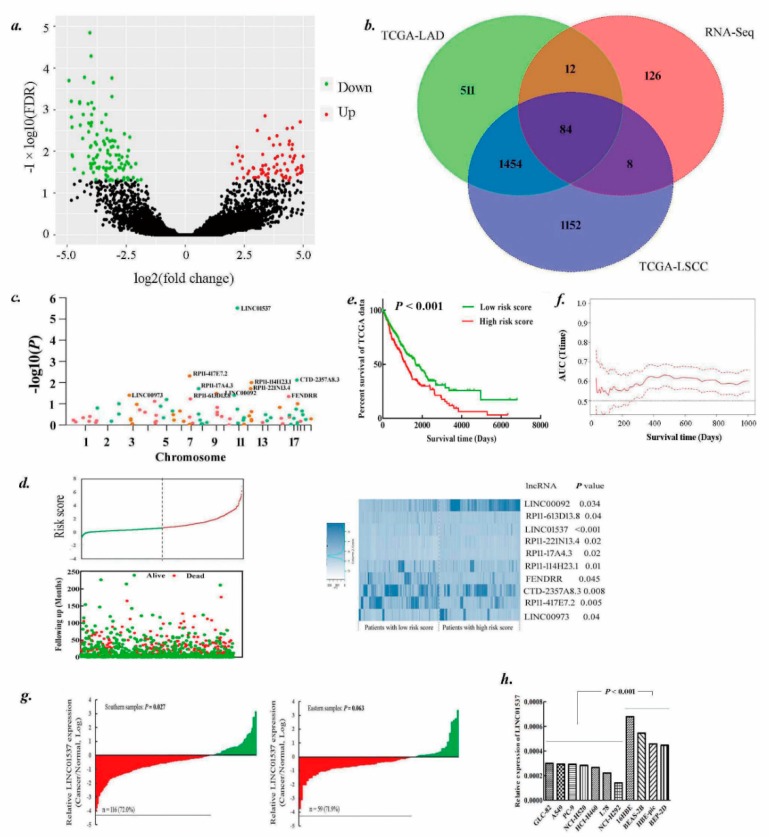
Identification of lung cancer survival-related long non-coding RNAs (lncRNAs). (**a**)**.** Volcano plot for comparison of lncRNAs expression profiles between eight pairs of lung cancer tissues and adjacent normal lung tissues. The x-axis indicates the differential expression profiles, plotting the fold-change in a log scale; the y-axis indicates the statistical significance of difference in expression. Green color dots represent downregulated lncRNAs and red ones represent upregulated lncRNAs with false discovery rate (FDR) < 0.05. (**b**)**.** Venn diagram for comparison of differentially expressed lncRNA profiles among RNA sequencing (RNA-seq) data, TCGA LUAD data, and squamous cell carcinoma (LSCC) data. (**c**)**.** Scatter plot of *p* values in –log10 scale from the univariate Cox model analysis on 84 lncRNAs that were overlapping in the above-mentioned differential lncRNA expression profiles. Green for downregulated lncRNAs and red for upregulated ones. Ten lncRNAs with significant *p* values were labeled. (**d**)**.** The distribution of ten-lncRNA risk score, patients’ survival status, and lncRNA expression. *p* value from the univariate Cox model. (**e**)**.** The Kaplan–Meier plot was used to visualize the survival probabilities for the low-risk versus high-risk group of TCGA lung cancer patients determined on the basis of the median risk score. The differences between the two curves were determined by the log-rank test. (**f**)**.** Prognostic score in the TCGA lung cancer cohort, predicted by the ten-lncRNA risk score in terms of the AUC (Area Under Curve.) of time-dependent receiver operating characteristic (ROC). The full line and dashed lines refer to AUC values and their 95% confidence intervals. (**g**)**.** The qPCR was performed to assess the expression of LINC01537 in the tissues of southern (left) and eastern (right) groups. *p* was calculated by the paired *t* test. (**h**) The qPCR was performed to assess LINC01537 expression in lung cancerous and immortalized normal cells. *p* was calculated by the Student’s *t* test.

**Figure 2 ijms-20-03713-f002:**
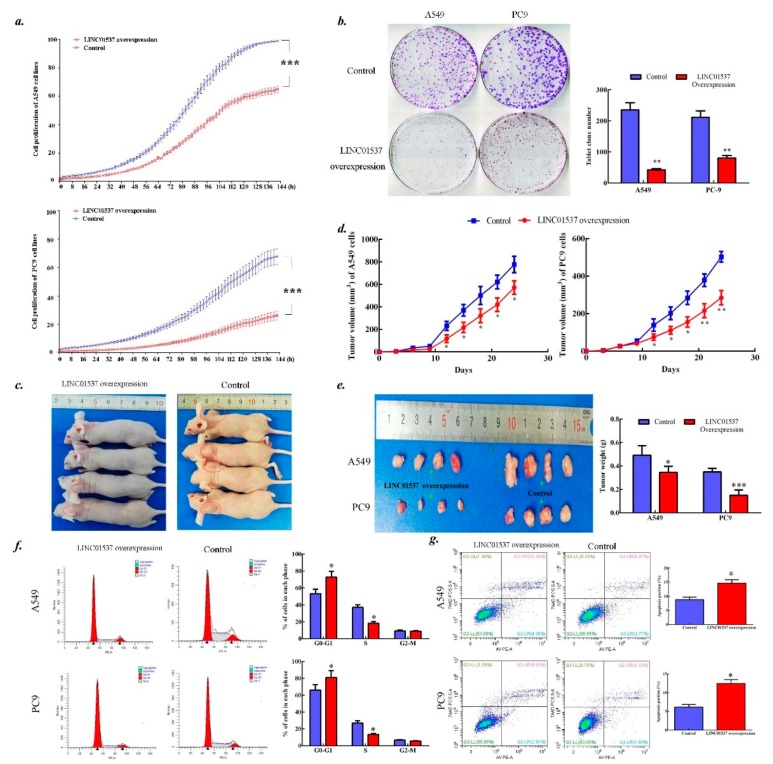
Biologic effects of LINC01537 on cell growth, cell cycle, and apoptosis. Both A549 and PC9 cells were transfected with pEZ-Lv201-LINC01537 (LINC01537 overexpression) and pEZ-Lv201-Empty (Control). (**a**). Cell proliferation was directly determined using the IncuCyte^®^ Live Cell Analysis Imaging System. (**b**). The plate colony assay was conducted to test for tumorigenicity. (**c**–**e**). The BALB/c nude mice were used to determine the growth of tumor in vivo, originating injection of A549 and PC9 cells. Tumor volume was measured every three days, until the end of third week. Extracted tumor tissues from the mice were further imaged and weighted. (**f**–**g**). Flow cytometry was conducted to determine the cell cycle (**f**) and cell apoptosis (**g**). Circle dot on line (**a, d**) or bar height (**b, e, f, g**) corresponds to the mean, and error bars represent the SD for five biological replicates, except for mice model with four replicates. **p* < 0.05, ***p* < 0.01, ****p* < 0.001, calculated by the Student’s t test.

**Figure 3 ijms-20-03713-f003:**
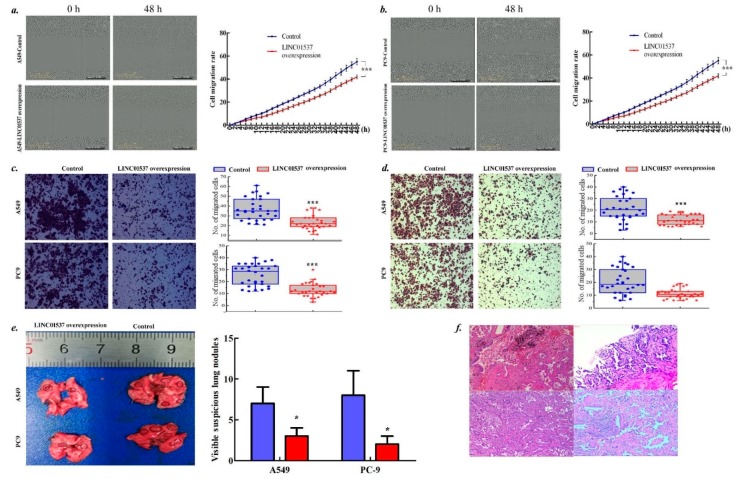
Biologic effects of LINC01537 on cell metastasis. Both A549 and PC9 cells were transfected with pEZ-Lv201-LINC01537 (LINC overexpression) and pEZ-Lv201-Empty (Control). (**a**–**b**). The wound healing was used to test cell migration in A549 cells (**a**) and PC9 cells (**b**) *in vitro*. (**c**–**d**). Transwell assay was applied for determining cell migration (**b**) and invasion (**c**) *in vitro*. (**e**). The tail vein infection experiment determined lung nodule *in vivo*. ***f.*** The IHC (Immunohistochemistry) confirmed the nodule harboring LUAD clinicopathological features. Circle dot on line (**a**) or bar height (**c**–**e**) corresponds to the mean, and error bars represent the SD for five (**a, b**), three (**c**–**e**) biological replicates. **p* < 0.05, ***p* < 0.01, ****p* < 0.001, calculated by the Student’s *t* test.

**Figure 4 ijms-20-03713-f004:**
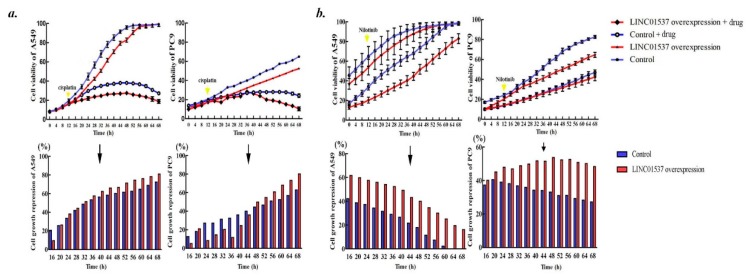
Biological effects of LINC01537 on chemosensitivity. Cell viability was directly determined using the IncuCyte^®^ Live Cell Analysis Imaging System. (**a**)**.** Cisplatin was added to the cell culture medium. (**b**)**.** Nilotinib was added. Circle dot on line (Up) or bar height (Down) corresponds to the mean, and error bars represent the SD for five biological replicates. Cell growth repression = OD(cells without Drug treatment)-OD(cells with Drug treatment)/OD(cells without Drug treatment).

**Figure 5 ijms-20-03713-f005:**
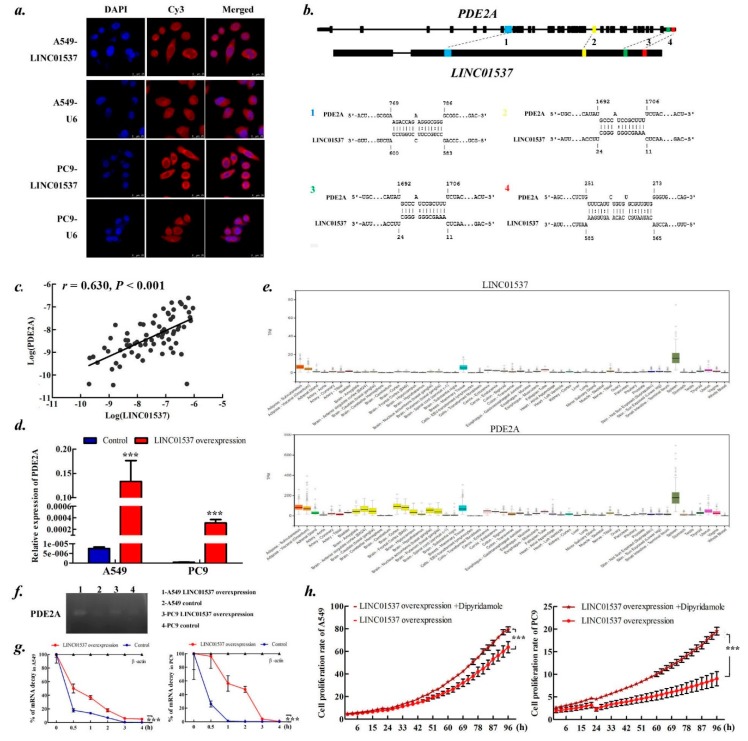
LINC01537 upregulates PDE2A via RNA–RNA interaction. (**a**)**.** Fluorescence in situ hybridization of LINC01537 in A549 and PC9 cells. DAPI staining of DNA (left), fluorescence in situ hybridization (FISH) probes of LINC01537 (middle) and merged (right). The small nuclear RNA U6 was tested as a positive control for nuclear localization. (**b**)**.** Schematic representation of the predicted RNA–RNA interaction between LINC01537 and the exons of PDE2A. (**c**)**.** The correlation between LINC01537 expression and PDE2A expression in lung cancer tissues. Purple dot corresponds to log (LINC01537) expression in x-axis and log(PDE2A) expression in y-axis. *r, p*, calculated by the Pearson correlation analysis. (**d**)**.** The qPCR was performed to determine PDE2A expression. (**e**)**.** LINC01537 and PDE2A expression in 53 tissues from GTEx RNA-seq. (**f, g**)**.** The RPA (f) and mRNA stability tests (**g**) were used to assess the role of LINC01537 on protecting PDE2A mRNA. The β-actin was tested as a control. ***h.*** Cell proliferation in response to dipyridamole treatment. Bar height (**d**) or Circle dot (**g, h**) on line corresponds to the mean, and error bars represent the SD for five biological replicates. ****p* < 0.001, calculated by the two-way ANOVA test.

**Figure 6 ijms-20-03713-f006:**
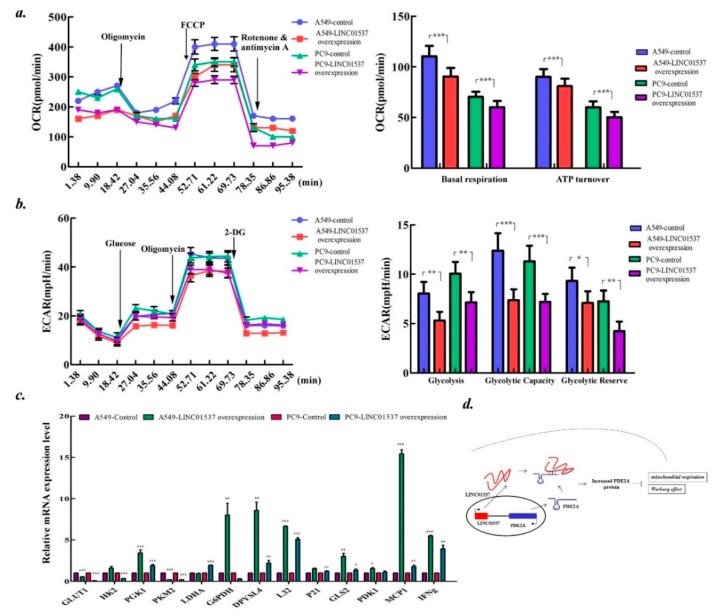
LINC01537 overexpression leads to attenuated Warburg effect and mitochondrial respiration. ***a-b.*** The Seahorse XF Glycolysis Stress Test Kit was used to determine oxygen consumption rate (OCR) (**a**) and extracellular acidification rate (ECAR) (**b**). (**c**)**.** The diagram shows the genes involve tumor energy metabolism which are induced by LINC01537 overexpression. ***d.*** A model depicts the regulatory mechanism of LINC01537 on PDE2A and energy metabolism. Bar height on line corresponds to the mean, and error bars represent the SD for five (a, b) or three (**c**) biological replicates.

**Table 1 ijms-20-03713-t001:** List of genes whose expression were significantly correlated with LINC01537 expression.

Gene Symbol	Lung Adenocarcinoma		Lung Squamous Cell Carcinoma	
	Correlation Coefficient	*p* Value	Correlation Coefficient	*p* Value
*PDE2A*	0.636	2.14 × 10^−41^	0.688	3.07 × 10^−32^
*CXorf36*	0.5	1.12 × 10^−23^	0.563	8.78 × 10^−20^
*PEAR1*	0.467	1.55 × 10^−20^	0.525	5.37 × 10^−17^
*RHOJ*	0.454	2.38 × 10^−19^	0.515	2.48 × 10^−16^
*LAMA4*	0.452	3.34 × 10^−19^	0.494	6.33 × 10^−15^
*TIE1*	0.451	4.46 × 10^−19^	0.481	3.82 × 10^−14^
*CD93*	0.448	8.47 × 10^−19^	0.508	7.57 × 10^−16^
*FLT4*	0.446	1.27 × 10^−18^	0.402	5.7 × 10^−10^
*MYCT1*	0.441	3.06 × 10^−18^	0.459	6.91 × 10^−13^
*MMRN2*	0.438	5.20 × 10^−18^	0.522	9.02 × 10^−17^
*VGLL3*	0.437	6.24 × 10^−18^	0.433	1.87 × 10^−11^
*GIPC3*	0.432	1.60 × 10^−17^	0.497	3.95 × 10^−15^
*GPR124*	0.432	1.89 × 10^−17^	0.405	4.21 × 10^−10^
*BNC2*	0.431	2.31 × 10^−17^	0.414	1.63 × 10^−10^
*LDB2*	0.431	2.27 × 10^−17^	0.438	9.76 × 10^−12^
*CD34*	0.43	2.42 × 10^−17^	0.476	7.32 × 10^−14^
*ROBO4*	0.43	2.60 × 10^−17^	0.513	3.77 × 10^−16^
*SYNPO*	0.428	3.99 × 10^−17^	0.486	2 × 10^−14^
*ELTD1*	0.422	1.04 × 10^−16^	0.489	1.19 × 10^−14^
*BCL6B*	0.417	2.98 × 10^−16^	0.465	3.14 × 10^−13^
*TIMP3*	0.411	8.14 × 10^−16^	0.421	7.16 × 10^−11^
*PDGFRB*	0.41	1.02 × 10^−15^	0.435	1.39 × 10^−11^
*ITGA5*	0.409	1.05 × 10^−15^	0.517	2.03 × 10^−16^
*SOX18*	0.409	1.16 × 10^−15^	0.421	7.41 × 10^−11^
*ARHGEF15*	0.408	1.33 × 10^−15^	0.438	9.87 × 10^−12^
*SPON1*	0.408	1.31 × 10^−15^	0.418	1.01 × 10^−10^
*CACNA1C*	0.406	1.87 × 10^−15^	0.42	8.64 × 10^−11^
*PALMD*	0.405	2.35 × 10^−15^	0.411	2.26 × 10^−10^
*MMP2*	0.401	4.86 × 10^−15^	0.414	1.62 × 10^−10^

**Table 2 ijms-20-03713-t002:** Demographics and clinical features of studied patients.

Characteristic	Southern Samples N (%)	Eastern Samples N (%)	Pearson χ^2^	*p* Value
Total	161 (66.3)	82 (33.7)		
Age				
<60	92 (57.1)	43 (52.4)	0.487	0.485
≥60	69 (42.9)	39(47.6)		
Gender				
Female	49 (30.4)	23 (28.0)	0.148	0.700
Male	112 (69.6)	59 (72.0)		
Family tumor history				
No	141 (87.6)	73 (89.0)	0.108	0.742
Yes	20 (12.4)	9 (11.0)		
Smoking				
No	62 (38.5)	26 (31.7)	1.088	0.297
Yes	99 (61.5)	56 (68.3)		
Stages				
I + II	61(37.9)	25(30.5)	1.301	0.254
III + IV	100(62.1)	57(69.5)		
Pathology				
Adenocarcinoma	77(47.8)	35(42.7)	0.899	0.638
Squamous carcinoma	44(27.3)	27(32.9)		
Other types *^a^*	40(24.8)	20(24.4)		

*^a^* Large cell carcinoma, small cell carcinoma, and hybrid or undifferentiated carcinoma.

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
