# Peer review of "Analysis of Survival-Related lncRNA Landscape Identifies A Role for LINC01537 in Energy Metabolism and Lung Cancer Progression"

_ijms, 2019, doi:10.3390/ijms20153713_

Reviewer 1 Report

This manuscript examines changes effects of lncRNAs in lung cancer. Overall the data are very strong and it is a good study. I have two comments. The manuscript has numerous grammatical errors that should be addressed. Also, the methods for the animal studies are not mentioned in the main text. They are provided in the supplementary methods but they should also be noted in the main text, even if only to specify that the details are given in the supplementary methods document.

Author Response

1. The manuscript has numerous grammatical errors that should be addressed.

      Response: We deeply regret that there were problems with the English. The paper has been carefully revised by a professional to improve the grammar and readability.

 2. Also, the methods for the animal studies are not mentioned in the main text. They are provided in the supplementary methods but they should also be noted in the main text, even if only to specify that the details are given in the supplementary methods document.

      Response: Thanks for bringing this to our attention. In response to this comment, we removed the description on animal experiments from the supplementary methods to the main text in the revision as follows: “

2.8 Tumor xenografts and neoplasms metastatic model in mice

Four to six-week-old female BALB/C-nu mice purchased from Huarongkang Biotechnology Co. Ltd. (Beijing, China) were raised at the Animal Experimental Center of Guangzhou Medical University. The mice were given a suspension of 2 × 106 cells subcutaneously. The mice were examined 3 times a week for 3 weeks and tumor growth was assessed by measuring length and width of the tumor mass. The volume of the tumor was calculated using the following formula: volume = length × width2 × 0.5. Each group included four mice. In other side, mice were given a tail vein injection of 2 × 106 cells and killed in six weeks after inoculation. The lungs were removed and fixed in 4% paraformaldehyde. The lung sections were stained with hematoxylin and eosin stain (HE) for histological examination. HE staining was performed following the routine protocol. Each group included four mice. All experiments and procedures involving animals were conducted in accordance with guidelines approved by the Laboratory Animal Center of Guangzhou Medical University.” 

Reviewer 2 Report

Dear Authors,

please find my comments here,

The authors have identified 230 lncRNAs including 110 up and 120 down) while 164 lncRNAs expressions were significantly different in lung cancer tissues and normal lung tissues.  Among that 10 lncRNAs were found to be significantly associated with lung cancer survival and the LINC01537 was the most significant one. 

1. the figures quality is really very poor. would you increase the pixels to get high or better resolution to atleast the words in the figures?

2. whats the difference in 164 lncRNAs and 10 lncRNAs ? i didnt understand what is the meaning of significantly different (164 lncRNAsand significantly associated (10 lncRNAs). would you explain it in more detail?

3. what is the role of 110 up-regulated lncRNAs in lung cancer? did you check functional role of those ? 

4. there are 120 down-regulated lncRNAs, i can understand you pick the top down-regulated for the functional characterization, however, it is good practice to pick few more from the list and test for functional characterization. would you explain? 

5. why not test other significantly associated lncRNAs? i agree you choose the one which is significantly higher, however we cant rule out other 9 lncRNAs which are also significantly higher. 

6. why didnt check the correlation of LINC01537 with other mRNAs like CXorf36, PEAR1, RHOJ etc,.(those p values are also statistically significant)

7. the mode of action of anti-cancer drug Nilotinib have be increased in over-expression system with LINC01537. is this true for other anti-cancer drugs ? if yes, i feel it will have some effects while considering to add as an adjuvant for the treatment. would you explain? 

Thanks,

LR

Author Response

1. The figures quality is really very poor. Would you increase the pixels to get high or better resolution to at least the words in the figures?

      Response: We deeply regret that there were problems with the figures.  We originally embedded all figures into the main text, which greatly damaged all figures’ quality. In the revision, we resubmitted all figures independently as figures.zip.  

2. What’s the difference in 164 lncRNAs and 10 lncRNAs? I didn’t understand what is the meaning of significantly different (164 lncRNAs) and significantly associated (10 lncRNAs). Would you explain it in more detail?

      Response: Thanks for the review’s comment. Just as mentioned in the paper, we aimed to find lncRNAs that were associated with lung cancer survival. As we implemented our strategy, we have performed RNA-seq analysis to reveal dysregulated lncRNAs in lung cancer by means of comparing the expression levels of them between lung cancer tissues and corresponding normal lung tissues. Based on this, we found 230 lncRNAs (not 164) with significant differences in expression between lung cancer tissues and corresponding normal lung tissues. Furthermore, we analyzed the correlation of these lncRNAs with survival of TCGA lung cancer patients and ultimately identified the 10 lncRNAs, which were significantly correlated with lung cancer survival. Therefore, significantly different means that the expression of the lncRNAs were statistically different between lung cancer tissues and corresponding normal lung tissues, while significantly associated defines that the lncRNAs were statistically correlated with lung cancer survival.      

3. What is the role of 110 up-regulated lncRNAs in lung cancer? Did you check functional role of those? 

      Response: Thanks for the review’s comment. Based on published literatures, there are rare lncRNAs of the 110 up-regulated ones such as UCA1 that have been studied. However, the roles of most lncRNAs are unclear. Since the number is big, we haven't really tried that hard to figure out their functional roles in lung cancer development. However, we will study them later.

4. There are 120 down-regulated lncRNAs, I can understand you pick the top down-regulated for the functional characterization, however, it is good practice to pick few more from the list and test for functional characterization. Would you explain? 

      Response: Thanks for bringing this to our attention. In fact, some of these lncRNAs are now being studied. However, on the account of that there are no evidence showing correlations between these lncRNAs and LINC01537, it is not appropriate to show data regarding other lncRNAs in the current paper.  

5. Why not test other significantly associated lncRNAs? I agree you choose the one which is significantly higher, however we can’t rule out other 9 lncRNAs which are also significantly higher. 

      Response: We agree with the review. In fact, the work on other lncRNAs are ongoing. Just as we mentioned above, there are no evidence showing correlations between these lncRNAs and LINC01537, it is not appropriate to show functional data regarding other lncRNAs in the current paper. LINC01537 is the first lncRNA we reported, and follow-up lncRNAs will be reported in future.   

6. Why didn’t check the correlation of LINC01537 with other mRNAs like CXorf36, PEAR1, RHOJ etc,.(those p values are also statistically significant).

      Response: Thanks for the review’s comment. Because a lot of studies showed that the regulation of cytoplasmic lncRNAs on target coding gene may be regulated by spatial position effect. We chose PDE2A not only because it ranked the first correlated gene, but also their chromosome locations are close. As shown by a lot of studies, the chromosomal locations of lncRNAs infer their biological functions and possibly target proteins. Furthermore, there is a great possibility that those other mRNAs are not direct targets of LINC01537, but just a butterfly effect caused by LINC01537.

7. The mode of action of anti-cancer drug Nilotinib have be increased in over-expression system with LINC01537. Is this true for other anti-cancer drugs? If yes, I feel it will have some effects while considering to add as an adjuvant for the treatment. Would you explain? 

      Response: Thanks for the review’s comment. Just as mentioned in our paper, we used two anti-cancer drugs but only found that LINC01537 augmented the anti-cancer effect of Nilotinib. We did not test other drugs. We discussed this result and thought that targeting LINC01537 may provide an additional option to certain patients.